# SCOPE: SPATIALLY-CONSTRAINED PARAMETRIC EDITING FOR TEXT-GUIDED CAD MODELS

## ABSTRACT

Text-based CAD editing (e.g., CAD-Editor) has emerged as a promising approach for automating CAD modifications from natural language instructions. However, existing methods lack explicit spatial understanding, limiting their ability to accurately interpret instructions that involve relative positions or geometric constraints. To address this gap, we introduce SCOPE, an extension of the locate-then-infill framework that integrates language-guided spatial reasoning into text-guided CAD editing. SCOPE enhances training by synthesizing spatially grounded editing samples and enables the model to learn spatial relations between CAD features (e.g., "drill a hole on the left panel above the rectangle"). Furthermore, we integrate spatial context into both the locate and infill stages, improving target region identification and spatially consistent modifications. Experiments on a public CAD dataset demonstrate that incorporating spatial reasoning significantly improves the accuracy of text-based CAD editing with more precise region localization and control while retaining the efficiency of the original framework.

## 1 INTRODUCTION

Computer-Aided Design (CAD) has become indispensable across modern engineering and manufacturing workflows where designers create complex models through iterative sketching and extrusion processes ( Li et al. (2023); Ren et al. (2022)). As CAD designing involves understanding **spatial relationship** between geometric components, designers constantly reference relative positions, orientations, and geometric constraints—language instructions in professional workflows frequently contain spatial context. Despite substantial progress in automating CAD generation, current methods either synthesize new models without user control ( Wu et al. (2021)) or support limited controllability through disentangled codebooks for topology and geometry ( Xu et al. (2022; 2023)), limiting fine-grained edits.

Text-based CAD editing aims to overcome these limitations by allowing designers to modify existing models via natural language commands. Early work on design variation generation yields uncontrolled random perturbations ( Wu et al. (2021); Xu et al. (2022)), while emerging text-to-CAD methods map descriptions to new models without leveraging prior design context ( Khan et al. (2024b); Li et al. (2025)). However, these approaches do not incorporate explicit spatial reasoning, a critical capability for interpreting instructions referring to relative positions such as "above," "next to," or "within" CAD features. Large Language Models (LLMs) and Large Vision-Language Models (LVLMs) have shown promise for sequence-to-sequence tasks and data synthesis across domains, including CAD code understanding( Xu et al. (2024); Wu et al. (2024). Usage of 2D CAD drawings Wang et al. (2025c) enabled 2D-to-3D parametric generation and editability across geometric entities and contexts. Yet existing pipelines do not generate spatially-grounded editing, preventing models from learning the geometric relationships fundamental to professional design workflows. This gap manifests in poor region localization and imprecise edit execution when spatial cues are present.

A core challenge in text-based CAD editing is successfully converting spatially descriptive instructions (e.g., "add a slot above the handle" or "align the chamfered edge next to the circular hole") into accurate parametric modifications. Flexible editing thus requires joint reasoning over spatial cues and underlying 3D geometry to enable precise identification and manipulation of target regions. However, most traditional methods either treat spatial cues implicitly or rely on global model repre-

sentations, lacking mechanisms that fuse spatial grounding with instruction parsing Li et al. (2023); Wang et al. (2025b); Zhang et al. (2025b); Wang et al. (2025a). As a result, models often misinterpret spatially anchored commands, which lead to ambiguities or incorrect editing outcomes Wang et al. (2025b); Zhang et al. (2025b). Bridging this gap requires frameworks that synthesize spatially explicit training data and reason jointly over language, geometry, and design context. This can support robust and controllable editing grounded in spatial semantics—an area that remains largely unexplored in current CAD-editing research Wang et al. (2025a); Khan et al. (2025).

In this work, we present **SCOPE**, a Spatially-Constrained Parametric Editing framework that fills this gap by (1) generating an open-source dataset of triplets—original CAD, spatially grounded instruction and edited CAD through an automated pipeline, and (2) extending the locate-then-infill paradigm with hierarchy-aware spatial tokens that explicitly model relative relationships between CAD features. SCOPE achieves significant gains in region localization and edit precision on public benchmarks, establishing a new standard for spatially-aware text-guided CAD editing. Our main contributions are:

- We propose **SCOPE**, a spatial context-aware parametric CAD editing framework that extends the locate-then-infill approach, enabling precise and bidirectional (forward and inverse) CAD modifications from natural language instructions.

- We propose an automated data synthesis pipeline to procedurally generate a large-scale spatially-grounded CAD editing dataset. To our knowledge, this is the first approach addressing the lack of spatial understanding in existing text-based CAD editing methods.

- Our method establishes a new state-of-the-art on the CAD editing benchmark, achieving significant gains over existing baselines. Our approach yields a 63.6% increase in the D-CLIP score, achieving significant improvements in both region localization and instruction-following for complex CAD editing tasks.

## 2 RELATED WORKS

**Large Language Models for CAD Generation.** Recent advances in Text-to-CAD research distinguished between CAD *generation* and *editing* tasks, where generation creates new designs from scratch and editing modifies generated models based on user instruction Khan et al. (2024b). CAD editing enables users to iteratively refine and customize models, offering flexibility that generation frameworks cannot offer. Large Language Models (LLMs) have transformed the text-to-CAD landscape. LLM-based CAD generation approaches ( Khan et al. (2024b); Li et al. (2024a; 2025)) focused on mapping textual prompts to new CAD models, often using two-tuple data (instruction, output model) and lack the capacity to interpret editing intent and preserve prior design context. However, Yuan et al. (2025) showed text-guided CAD editing requires triplet structures (instruction, original model, edited model) to enable step-by-step, spatial-aware, and user-driven manipulations, which is not addressed by LLM-based generation works Zhang et al. (2025c). Multimodal approaches ( Xu et al. (2024); Wu et al. (2024); Alrashedy et al. (2025)) leveraged LVLMs with image input to enrich generative capacities, but failed to enable an editing step for reuse and human-centric design iteration. Unlike previous strategies that rely on direct text-to-object mapping or require extensive user segmentation input ( Li et al. (2020; 2022)), integrating LLMs/LVLMs into editing frameworks enables automated and flexible operation over both global and local model hierarchies, and supports summarizing the difference between model states—critical for effective edit tracking and guidance ( Khan et al. (2024b)). Stepwise captioning and context-aware editing further differentiate editing-focused pipelines, allowing LLMs to facilitate fine-grained modifications while maintaining the design intent. This substantially makes the CAD design process more interactive and adaptable.

**Sequence modeling for CAD.** Transformer-based models ( Vaswani et al. (2017)) inspired CAD construction as autoregressive modeling. Recent frameworks utilize domain-specific languages (DSL) to represent geometric operations as discrete tokens, enabling autoregressive generation of CAD sequences ( Willis et al. (2021); Ganin et al. (2021)). DeepCAD ( Wu et al. (2021)) pioneered transformer-based autoencoders for sketch-and-extrude modeling and more subsequent works developed for sequence representations ( Seff et al. (2022); Xu et al. (2022); Guo et al. (2022); Xu et al. (2023)). Building upon this sequential modeling approach, our approach incorporates spatial

reasoning to understand and execute editing instructions on relative positions and geometric relationships between CAD components.

**Spatially-grounded CAD editing.** While recent advances in Text-to-CAD editing have demonstrated significant progress across different representation paradigms, existing approaches predominantly focus on shape synthesis ( Zhang et al. (2025c); Govindarajan et al. (2025)) without incorporating explicit spatial reasoning capabilities. Current text-based CAD editing methods ( Khan et al. (2024a); Li et al. (2024b)) can interpret basic geometric descriptions but struggle with instructions containing spatial relationships between CAD components. For instance, unconditional generation approaches like DeepCAD ( Wu et al. (2021)) lack any form of spatial control, while controllable methods such as SkexGen ( Xu et al. (2022)) and Hnc-cad ( Xu et al. (2023)) provide topology and geometry control but cannot handle relative positioning constraints. This limitation becomes particularly evident in CAD editing scenarios where users frequently reference spatial relationships (e.g., "add a hole above the existing rectangle" or "extend the left panel"). The absence of spatial understanding requires post-processing, significantly limiting the practical utility of automated CAD manipulation systems. Moreover, existing training pipelines lack spatial grounding( Yuan et al. (2025)), limiting models from learning the spatial relationships fundamental to professional CAD workflows Wang et al. (2025b). This gap highlights the need for CAD frameworks that can handle spatial relationships for intuitive parametric CAD design. In contrast to these approaches, our method explicitly integrates spatial reasoning into both the training data synthesis and model architecture, which offers relative positioning controls during editing operations.

## 3 METHODOLOGY

### 3.1 SCOPE FRAMEWORK

In this section, we introduce SCOPE, a novel framework for text-guided CAD editing that explicitly models spatial understanding to interpret the edit instructions. As illustrated in Figure 1, SCOPE adopts two finetuning stages (locate and infill) from CAD-Editor( Yuan et al. (2025)), where it finetunes an LLM for CAD editing with **spatial context**. In the locate stage, it is finetuned with the original CAD sequence and the editing instruction tokens to predict a masked sequence where a target region is replaced by a <mask> token. The target region is a specific part of the token sequence derived from the CAD manipulation instruction. Subsequently, the infill stage uses this masked sequence along with the two inputs from the locate stage to generate the final edited CAD sequence by filling in the <mask> tokens with generated tokens. SCOPE addresses the lack of explicit spatial understanding in CAD-Editor's finetuning strategy by incorporating spatially-grounded data and a dedicated spatial relation <SR> token into the finetuning pipeline. The <SR> token guides both stages to achieve precise, context-aware region localization and consistent CAD modification.

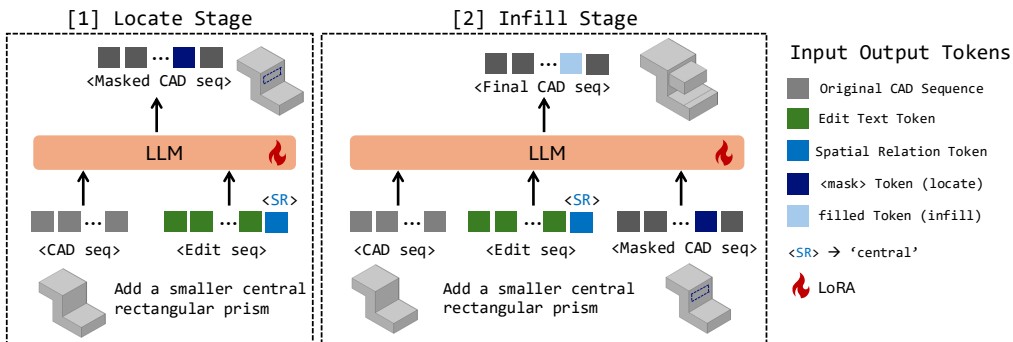

Figure 1: **Overview of the SCOPE framework.** The framework consists of LLM finetuning in a two-stage locate-then-infill Yuan et al. (2025). The locate stage predicts <mask> tokens in the original CAD sequence based on spatially-augmented (spatial relation <SR> token) editing instructions. The infill stage translates masked sequences and spatial cues into accurately edited CAD models with proper spatial grounding.

## 3.2 Spatially-Constrained CAD Editing

We formulate text-guided CAD modification as a conditional sequence generation task. Let $\mathcal{I}$ denote the language command for CAD editing, $\mathcal{C}_{\text{main}}$ be the token sequence of the original CAD model and $\mathcal{C}_{\text{edit}}$ be the target edited sequence. A parameterized function $f_\theta$ learns to map the instruction and main model to the desired edit:

$$\mathcal{C}_{\text{edit}} = f_\theta(\mathcal{I}, \mathcal{C}_{\text{main}}) \tag{1}$$

This is typically optimized by maximizing the log-likelihood of the target sequence over a dataset $\mathcal{D}$ of $(\mathcal{I}, \mathcal{C}_{\text{main}}, \mathcal{C}_{\text{edit}})$ triplets:

$$\mathcal{L}(\theta) = \sum_{(\mathcal{I}, \mathcal{C}_{\text{main}}, \mathcal{C}_{\text{edit}}) \in \mathcal{D}} \log P_\theta(\mathcal{C}_{\text{edit}} \mid \mathcal{I}, \mathcal{C}_{\text{main}}) \tag{2}$$

A major challenge in this formulation is the model's ability to interpret spatial context within $\mathcal{I}$ to accurately localize edits. Standard LLMs often exhibit "bag-of-tokens" behavior ( Qi et al. (2025)), struggling to ground spatial relations like "above" or "infront of" within the geometric context of $\mathcal{C}_{\text{main}}$. This leads to ambiguity in the conditional probability distribution over $\mathcal{C}_{\text{edit}}$, given $\mathcal{I}$ and $\mathcal{C}_{\text{main}}$. To address this, SCOPE extends the problem formulation by introducing an explicit spatial relation <SR> token into the vocabulary. The input edit instruction $\mathcal{I}$ is augmented to $\mathcal{I}' = (\mathcal{I}, \text{<SR>})$ whenever spatial constraints are present. This token provides a strong inductive bias, enabling the model to explicitly condition its predictions on spatial context. Thus, the locate-then-infill stages are refined as conditional generation with spatial constraints. Let $\Omega$ be the set of token indices whose semantics are spatially dependent, then 1 extends to:

$$P_\theta(\mathcal{C}_{\text{edit}} \mid \mathcal{I}, \mathcal{C}_{\text{main}}) \approx P_\theta(\mathcal{C}_{\text{edit},\Omega} \mid \mathcal{I}', \mathcal{C}_{\text{main}}) \cdot P_\theta(\mathcal{C}_{\text{edit},\bar{\Omega}} \mid \mathcal{I}, \mathcal{C}_{\text{main}}) \tag{3}$$

**1. Locate Stage:** The model learns to predict a masked sequence $\mathcal{C}_{\text{mask}}$ where tokens corresponding to the target region are replaced by a <mask> token. The <SR> sharpens the conditional distribution:

$$\mathcal{C}_{\text{mask}} \sim P_\theta(\cdot \mid \mathcal{I}', \mathcal{C}_{\text{main}}) \tag{4}$$

The presence of <SR> guides the model's attention mechanism to more efficiently identify the subsequence in $\mathcal{C}_{\text{main}}$ that corresponds to the spatial instruction, thereby improving localization accuracy.

**2. Infill Stage:** The model generates the final edited sequence $\mathcal{C}_{\text{edit}}$ by filling in the masked regions. This stage is conditioned on the full spatial context provided by the instruction, the original model, and the precisely localized <mask> tokens:

$$\mathcal{C}_{\text{edit}} \sim P_\theta(\cdot \mid \mathcal{I}', \mathcal{C}_{\text{main}}, \mathcal{C}_{\text{mask}}) \tag{5}$$

By explicitly tokenizing spatial relations, we transform an implicit reasoning challenge into a structured sequence modeling task. As supported by theoretical work on spatial reasoning in language models ( Wang et al. (2025b); Qi et al. (2025)), this factorization improves the optimization landscape. It allows the model to learn a more sample-efficient mapping from instruction to spatially precise design modifications. The joint optimization objective for SCOPE becomes:

$$\mathcal{L}_{\text{SCOPE}}(\theta) = \sum_{(\mathcal{I}', \mathcal{C}_{\text{orig}}, \mathcal{C}_{\text{edit}}) \in \mathcal{D}} [\log P_\theta(\mathcal{C}_{\text{mask}} \mid \mathcal{I}', \mathcal{C}_{\text{main}}) + \log P_\theta(\mathcal{C}_{\text{edit}} \mid \mathcal{I}', \mathcal{C}_{\text{main}}, \mathcal{C}_{\text{mask}})] \tag{6}$$

We observe that naive models (without <mask> tokens) implicitly treat all tokens as i.i.d. or rely on standard positional embeddings. However, Zhang et al. (2025a) shows that positional tokens alone are insufficient for nuanced spatial reasoning. Therefore, our approach ensures that both localization and modification are directly conditioned on the spatial semantics embedded within the instruction, where the <mask> token serves as a powerful anchor for the model's attention mechanism.

## 3.3 CAD as Structured Text Representation

Our approach builds upon the established sketch-and-extrude modeling (SEM) ( Wu et al. (2021); Xu et al. (2022)) while incorporating **spatial awareness** throughout the hierarchical construction process. CAD models naturally exhibit multiple construction hierarchies, from basic geometric

primitives to complete 3D bodies. At the foundational level, curves serve as elementary building blocks encompassing lines ($l$), arcs ($a$), and circles ($c$). Each of them is defined by specific coordinate points: lines by two endpoints, arcs by three control points, and circles by center and radius points. These curves aggregate into loops ($L$), which form closed geometric paths either as single entities (e.g., standalone circles ($c$)) or composite structures (e.g., line-arc-line sequences ($l$-$a$-$l$) that define complex contours). Faces ($F$) represent bounded 2D regions, characterized by outer boundary loops and potentially multiple inner loops that create holes or cutouts within the geometry. Sketches ($S$) comprise collections of faces that share common extrusion parameters and geometric constraints. Extrusions ($E$) transform 2D sketches into 3D volumes through operations like additive extrusion or subtractive cuts. Finally, complete CAD models ($M$) consist of multiple sketch-extrusion ($SE$) entities that collectively define the final parametric design.

We follow the FlexCAD ( Zhang et al. (2025c)) approach to convert CAD models into structured text sequences using specialized termination tokens ('$H_{end}$' for $H \in \{$curve, loop, face, sketch, extrusion$\}$) to separate out each hierarchy and enable precise manipulation. Spatial descriptors (e.g., "above," "left of") are interleaved with coordinate tokens so that, for an instruction like "drill a hole on the left panel above the existing rectangle," the model can directly identify and locate the target and reference components within the hierarchy. During the spatial-context aware fine-tuning, we pass standard CAD sequences with instruction-editing triplets—such as "add a cylindrical hole above the rectangular cutout on the front face" paired with before and after texts—to teach the model to predict curve-level, loop-level, and sketch-level modifications conditioned on spatial language. This hierarchy-aware masking strategy ensures the model learns to map relative spatial instructions to precise geometric transformations across all SEM levels for spatial-aware CAD edits.

### 3.4 SPATIALLY-GROUNDED DATA SYNTHESIS

To effectively train our model for spatially-aware CAD editing, we develop an automated pipeline to generate a large-scale, spatially-grounded dataset. This process is designed to create instruction-edit triplets, denoted as $\mathcal{D} = \{(I, \mathcal{C}_{\text{main}}, \mathcal{C}_{\text{edit}})\}$. We begin by filtering CAD models from the Deep-CAD dataset ( Wu et al. (2021)). For each model, we employ a design variation model, Hnc-CAD ( Xu et al. (2023)), to generate a diverse set of modified counterparts. This creates $(\mathcal{C}_{\text{main}}, \mathcal{C}_{\text{edit}})$ pairs that include a wide range of common CAD edit operations, including the addition, deletion, and modification of geometric features. Next, for each pair of CAD models, we generate a corresponding editing instruction that is both descriptive and spatially precise. This is achieved by leveraging a large vision-language model[1] (LVLM) to analyze rendered images of the original and edited models. While CAD-Editor also uses LVLMs to generate editing instructions, our approach is specifically designed to elicit and embed **spatial relation** in the instruction. We use a stepwise captioning strategy where the LVLM is prompted to perform a three-step iteration- first: describe the geometric properties of each model; second: identify the differences between them; and third: compress these differences into a concise editing instruction. The LVLM generates a natural language editing instruction ($\mathcal{I}$) with explicit spatial relationships (e.g., "above, "to the right of"), which are then encoded as dedicated spatial relation <SR> tokens.

Each generated triplet is structured to explicitly include this **spatial relation**, as illustrated in the following example for adding a prism to the `right` of an existing CAD model:

```
{
  "original_sequence": "line,9,9 <curve_end> ... <extrude_end>",
  "edited_sequence": "line,9,9 <curve_end> ...",
  "masked_sequence": "line,9,9 <curve_end> ... <mask>",
  "instruction": "Add a smaller triangular prism 7 units to the right.",
  "type": "add",
  "method": "sequence",
  "spatial_relation": "right"  ⟶ spatial context added
}
```

This explicit encoding of **spatial relation** differentiates our approach from CAD-Editor, which relies on a general Longest Common Subsequence (LCS) algorithm to create ground-truth masked sequences. While LCS can identify token-level differences, it is agnostic to the underlying spatial

---

[1]https://huggingface.co/openai/gpt-oss-20b

semantics of the edit. In contrast, our `<SR>` tokens provide a direct, unambiguous guidance signal to the model.

## 3.5 PROMPT TEMPLATE WITH SPATIALLY-GUIDED MASKING

Our method enhances the prompt template of CAD-Editor by incorporating the spatial context. During finetuning stages, we employ a hierarchy-aware masking strategy ( Xu et al. (2023)) where the `<SR>` token guides the model to the precise target region. The masking process is applied across different hierarchical levels: for example, an entire internal sketch-extrusion is replaced with a single `<sketch-extrusion mask>` for high-level edits, allowing the model to learn to generate structures of varying complexity. Unlike FlexCAD ( Zhang et al. (2025c)), our masking does not rely on structural difference but is semantically linked to the user's spatial intent. For instance, `<face mask>` or `<curve mask>` tokens are inserted at hierarchical levels directly corresponding to the region identified by the `<SR>` token. This approach allows the model to condition its predictions on precise spatial semantics during both the locate and infill stages. `<SR>` guided masking process allows SCOPE to robustly interpret and execute complex, spatially-constrained CAD edits during inference.

Our spatially-guided masking is implemented through a two-stage finetuning process (locate and infill) with distinct prompt templates for each stage. The prompts are structured as follows:

**Locate Stage Prompt Template**

```
Inst: Below is a CAD operation
    sequence. Replace the parts
    that need to be modified with
    the string <mask> according to
    the editing instruction.
Input: <Inst> <OrigSeq>
Output: <MaskedSeq>
```

**Infill Stage Prompt Template**

```
Inst: Based on the original CAD
    sequence, editing instruction
    and masked sequence, generate
    the edited CAD sequence that
    could replace <mask> in the CAD
     model.
Input: <Inst> <OrigSeq> <MaskedSeq>
Output: <EditedSeq>
```

The terms in the prompt templates define the data flow for each training stage. The `<Inst>` is the natural language instruction guiding the CAD edit. The `<OrigSeq>` refers to the token sequence of the original CAD model before modification. The `<MaskedSeq>` in the locate stage denotes the intermediate representation where the specific tokens of the target region are replaced by a `<mask>` token. The `<mask>` tokens are inserted at curve-, loop-, or face-level boundaries based on spatial descriptors (e.g., "above," "left of"), guiding the model to predict which $SE$ hierarchies require modification. Finally, the `<EditedSeq>` is the output sequence from the infill stage, representing the fully modified final CAD model with the new geometry generated in place of the masked tokens. We jointly optimize both stages using cross-entropy loss over the generated tokens. Given an editing instruction `<Inst>`, an original SE token sequence `<OrigSeq>`, we first train an LLM to predict a `<MaskedSeq>` by inserting `<mask>` tokens at hierarchical boundaries indicated by spatial descriptors (e.g., "right", "above"). Then, in the second stage, the model attends to the triplet—`<Inst>`, `<OrigSeq>`, and `<MaskedSeq>`—to generate the precise edited sequence `<EditedSeq>` while preserving the unmasked portions. This framework allows generating a targeted and spatially accurate CAD edit sequence.

## 4 EXPERIMENTS

### 4.1 EXPERIMENTAL SETUP

**Datasets.** We build upon the DeepCAD ( Wu et al. (2021)), comprising 178K semantically rich Sketch-and-Extrude models. After de-duplication following prior protocols ( Xu et al. (2022; 2023)), we allocate 90% for training, 5% for validation, and 5% for held-out testing. To furnish spatially grounded editing examples, our automated pipeline (Section 4) synthesizes 150K triplets—each containing an original model, a spatial instruction, and the corresponding edited model with relative-position descriptors ("`spatial_relation`") at all hierarchy levels. For evaluation, we randomly select 2000 cases from the held-out test split, generate initial edited versions via the same pipeline, and

perform manual verification to guarantee spatial-instruction fidelity. Each test instance is processed to produce three edit hypotheses, resulting in 6000 candidate outputs for comparative benchmarking.

**Models.** The spatial data synthesis leverages GPT-oss Agarwal et al. (2025) for visual-difference summarization and LLaMA-3-70B for sequence-level instruction generation. Fine-tuning employs Gemma3-1B Kamath et al. (2025) as the base, optimized for 70 epochs under PyTorch Distributed Data Parallel (DDP) on NVIDIA A100×80 GB GPUs. We initialize learning at $1 \times 10^{-4}$, Max token length at 1024, and integrate LoRA adapters of rank 16, batch size 2. At inference, we sample with a temperature of 0.8 and top-p of 0.9 to balance precision and diversity in generated edits.

**Metrics.** (1) Validity Ratio (VR): Proportion of generated CAD sequences that can be successfully parsed and rendered into 3D models.

$$\text{VR} = \frac{N_{\text{valid}}}{N_{\text{total}}} \tag{7}$$

(2) Jensen–Shannon Divergence (JSD) for realism: Measures similarity between point cloud distributions of generated and ground-truth models Wu et al. (2021). It measures KL divergences ( Kullback & Leibler (1951)) of $P$ and $Q$ from the mean distribution $M$.

$$\text{JSD}(P\|Q) = \frac{1}{2}\text{KL}(P\|M) + \frac{1}{2}\text{KL}(Q\|M) \tag{8}$$

$$M = \frac{1}{2}(P + Q) \tag{9}$$

(3) Chamfer Distance (CD): Measures the geometric distance between the edited and ground-truth CAD point sets. It calculates how close the point sets $(S_1, S_2)$ are by averaging the nearest neighbor distances.

$$\text{CD}(S_1, S_2) = \frac{1}{|S_1|} \sum_{x \in S_1} \min_{y \in S_2} \|x - y\|_2^2 + \frac{1}{|S_2|} \sum_{y \in S_2} \min_{x \in S_1} \|x - y\|_2^2 \tag{10}$$

(4) Directional CLIP Score (D-CLIP): Using the CLIP score ( Sohn et al. (2023)), D-CLIP assesses if the visual change made to a CAD model matches the semantic direction described by the text instruction. Both images and texts are encoded in CLIP's ( Radford et al. (2021)) multimodal space and D-CLIP computes the cosine similarity between the difference in image features and the difference in text features induced by the edit. The score consists of two directional embeddings: visual direction ($\Delta v$) and textual direction ($\Delta u$).

$$\Delta v = \mathcal{E}_{\text{img}}(I_{\text{edit}}) - \mathcal{E}_{\text{img}}(I_{\text{orig}}), \quad \Delta u = \mathcal{E}_{\text{text}}(t_{\text{edit}}) - \mathcal{E}_{\text{text}}(t_{\text{base}}) \tag{11}$$

$$\text{D-CLIP} = \frac{\langle \Delta v, \Delta u \rangle}{|\Delta v|_2 \cdot |\Delta u|_2} \tag{12}$$

Here, $\mathcal{E}_{\text{img}}$ and $\mathcal{E}_{\text{text}}$ are the CLIP image and text encoders, $I_{\text{orig}}$ and $I_{\text{edit}}$ denote the image of the original and edited CAD respectively, while $t_{\text{base}}$ and $t_{\text{edit}}$ are the baseline (e.g., "This is a 3D shape") and textual instruction prompts. The final D-CLIP score is computed as the cosine similarity between $\Delta u$ and $\Delta v$.

**Baselines.** We evaluate our approach against three baseline approaches- including those producing design variants (e.g., SkexGen Xu et al. (2022), Hnc-CAD Xu et al. (2023), FlexCAD Zhang et al. (2025c)), text-driven CAD synthesis (Text2CAD Khan et al. (2024b)), and large pretrained foundation models such as GPT-4o with both standard (GPT-4o-Basic) and in-context prompting (GPT-4o-IC). We report D-CLIP scores to quantitatively assess the fidelity of spatially-grounded modifications—emphasizing the alignment between user intent and visual edits. All methods are evaluated using standard metrics described above and reported in Section 4.2. Since baseline methods do not incorporate spatial reasoning, D-CLIP serves as a promising metric for evaluating how well CAD edits align with user-specified spatial instructions.

## 4.2 PERFORMANCE COMPARISON RESULTS

**Qualitative Comparison.** Our qualitative results demonstrate that SCOPE consistently outperforms both CAD-Editor and GPT-4o-IC in instruction following (specifically for the edits requiring spatial

understanding) and edit quality. As shown in Figure 2, while baseline models often struggle with spatial editing commands—such as placing an object, maintaining symmetrical alignment—SCOPE successfully interprets and executes them in a 3D context. For instance, when tasked with adding a feature (e.g., "centrally beneath the arch"), SCOPE correctly identifies the reference geometry and applies the edit with high precision. Similarly, for instructions requiring alignment with existing features (e.g., "aligning with the left hole"), it exhibits superior geometric fidelity. In contrast, CAD-Editor and GPT-4o-IC frequently fail to ground these spatial constraints, which result in misplaced geometry or incomplete edits. This trend highlights the effectiveness of our framework to translate nuanced user commands into accurate and spatially-aware CAD edits.

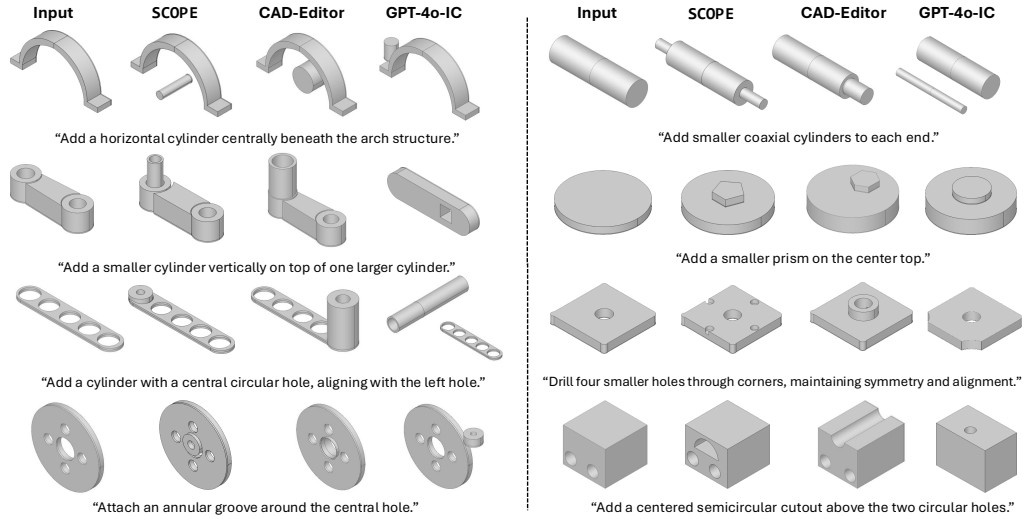

Figure 2: **Comparison of CAD editing with SOTA methods.** The input CAD model is on the left with the edited model on the right and the corresponding editing instruction is provided below. Compared to CAD-Editor and GPT-4o-IC, SCOPE consistently generates edits that closely align with the user's spatial intents, showcasing its improved localization precision and geometric fidelity.

**Quantitative Comparison.** As shown in Table 1, SCOPE demonstrates highly competitive performance against existing state-of-the-art methods. Even with a significantly smaller Gemma3-1B backbone, our model achieves a superior D-CLIP score (0.15) compared to CAD-Editor (0.11), underscoring the effectiveness of our spatially-grounded framework.

Table 1: **Comparison of SCOPE with CAD editing methods across different evaluation metrics.** Bold values indicate the best results and underlined values denote the second-best.

| Method | Backbone | VR ↑ | JSD ↓ | CD ↓ | D-CLIP ↑ |
|---|---|---|---|---|---|
| SkexGen | - | 74.3 | 1.94 | – | – |
| Hnc-CAD | - | 77.4 | 1.77 | – | – |
| FlexCAD | Llama3-8B | 82.1 | 1.72 | – | – |
| Text2CAD | Mistral-7B | 84.8 | 2.39 | 1.91 | – |
| GPT-4o-Basic | >1B | 63.2 | 1.10 | 2.30 | – |
| GPT-4o-IC | >1B | 84.5 | 0.70 | 1.55 | – |
| CAD-Editor | Llama3-8B | **95.6** | 0.65 | 1.18 | 0.11 |
| **SCOPE** (ours) | Gemma3-1B | 75.5 | 1.82 | 1.51 | 0.15 |
| **SCOPE** (ours) | Gemma3-12B | 91.3 | **0.61** | **1.12** | **0.18** |

When scaled to the Gemma3-12B backbone, SCOPE establishes a new SOTA, outperforming CAD-Editor in almost all key metrics. It improves upon CAD-Editor's realism with a 6.2% reduction in JSD, enhances geometric alignment with a 5.1% decrease in Chamfer Distance, and demonstrates a remarkable 63.6% increase in instruction-following capability as measured by the D-CLIP score while maintaining a high validity ratio of 91.3. These results confirm that embedding explicit spatial

relation <SR> tokens into the training pipeline significantly improves CAD editing performance. The substantial gains in the D-CLIP score, in particular, provide direct evidence that SCOPE more accurately interprets user spatial intent compared to other approaches, which lack this spatial guidance. The fact that our larger model surpasses the previous SOTA across all key metrics demonstrates that this architectural enhancement is not merely a trade-off but a fundamental improvement that scales effectively with model capacity, leading to superior geometric fidelity and instruction-following.

### 4.3 CAD EDITING WITH SPATIAL MANIPULATION

Spatial manipulation tasks reveal SCOPE's effectiveness in precise spatial editing. Figure 3 shows qualitative results highlighting its robust performance across diverse spatial editing scenarios. It showcases that it can execute challenging edits, including modifying sizes, shapes, and positions, as well as understanding multi-object relativity. Additionally, the model interprets parameterized text instructions with explicit numeric values (e.g., "Increase the central hole radius by 3 units"). Moreover, it successfully follows multiple spatial relations in complex prompts (e.g., "Add a smaller cylinder at the bottom right corner").

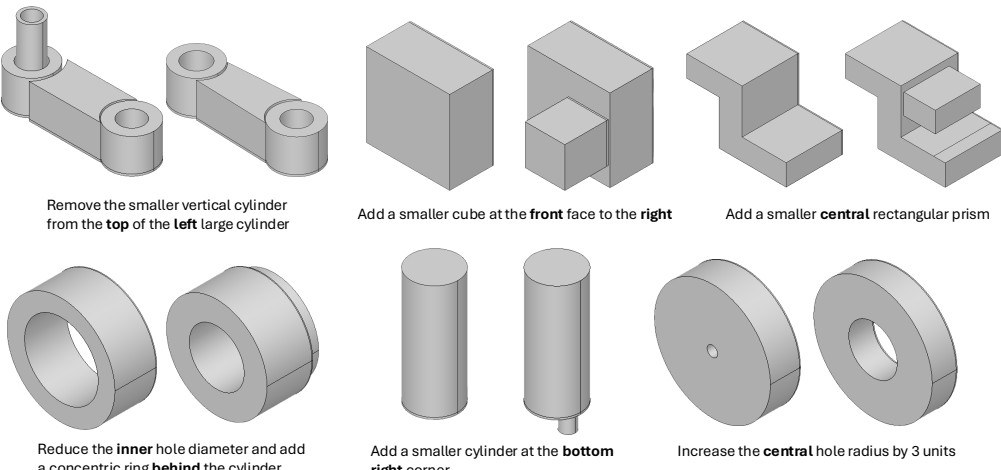

Remove the smaller vertical cylinder from the **top** of the **left** large cylinder

Add a smaller cube at the **front** face to the **right**

Add a smaller **central** rectangular prism

Reduce the **inner** hole diameter and add a concentric ring **behind** the cylinder

Add a smaller cylinder at the **bottom right** corner

Increase the **central** hole radius by 3 units

Figure 3: **Examples of Spatial CAD editing using SCOPE.** The left model indicates input, the right one is the output and the corresponding input instruction is shown below. The prompt highlights the spatial relation terms in bold. It shows SCOPE accurately interprets and applies spatial instructions— left, right, front, center, etc. to perform precise and spatially-grounded CAD edits.

## 5 CONCLUSION

We present SCOPE, a unique framework for text-based parametric CAD editing that integrates *spatial reasoning* into both data synthesis and sequence modeling. Our approach enhances the locate-then-infill method with hierarchy-aware *spatial tokens*, achieving superior performance in region localization and edit precision. We proposed an automated spatially-grounded data synthesis pipeline to address a critical training-data bottleneck and establish a robust benchmark for future research. SCOPE paves the way for more intuitive, context-aware CAD design workflows, empowering users to perform complex edits through natural language instructions.

While SCOPE marks a significant step forward, we acknowledge its limitations. First, the data synthesis pipeline's reliance on LVLMs presents a bottleneck for scalability. Second, the model's ability to interpret highly complex and compositionally dense spatial instructions remains an open challenge. Finally, the framework's performance is inherently tied to the quality of the synthesized data, where inaccuracies in generated instructions could propagate errors. Future work will focus on developing more efficient data synthesis methods and enhancing the model's compositional reasoning for multi-step spatial edits.

## REPRODUCIBILITY

Our code is shared at this anonymous link. We promise to open-source the code after acceptance. To promote reproducibility, we include detailed hyperparameter settings in Appendix 8.2.

## ETHICS STATEMENT

The data and methods presented in this work are specifically designed for the creation and modification of Computer-Aided Design (CAD) models. Given this specialized application, the risk of misuse is inherently low, ensuring that our research primarily benefits professional design and engineering disciplines. We have adhered to ethical guidelines in all aspects of our work. No human subjects, personal data or human evaluations were involved in this research. Our research is solely dependent on improving the language-guided CAD generation and modification. We strongly believe this research does not raise serious ethical concerns.

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

# Supplementary Material

## 6 REPRODUCIBILITY

Our code is shared at this anonymous link. We promise to open-source the code after acceptance. As part of the reproducibility, we include detailed hyperparameter settings in XX.

## 7 IMPACT STATEMENT

Our method introduces a novel framework for text-guided CAD editing that significantly improves a model's ability to understand and execute commands with complex spatial constraints. The presented work will facilitate future research into more sophisticated and spatially-aware generative models for engineering and manufacturing.

## 8 EXPERIMENT AND IMPLEMENTATION DETAILS

### 8.1 COMPUTATIONAL RESOURCES

All experiments are conducted on two NVIDIA A100 GPUs equipped with 80GB of memory. The implementation is based on PyTorch and leverages the Hugging Face Transformers library for model execution. For large language model runs, we adopt the official default system prompt and set the decoding temperature to 0.6, ensuring a trade-off between stability and diversity.

### 8.2 HYPERPARAMETERS

We fine-tune the Gemma3-1B and Gemma3-12B backbones using two NVIDIA A100 GPUs, each with 80GB memory. Training is performed with the **AdamW** optimizer Loshchilov & Hutter (2017) at a learning rate of $1 \times 10^{-4}$. We adopt the lowa rank (LoRA) finetuning with rank 16 for a parameter-efficient finetuning strategy. Training is conducted using mixed precision with bfloat16. The batch size is set to 2. We set a gradient accumulation with 4 steps and set the weight decay to 0.01 for all training runs. This yields an effective batch size of 8. We train the model for 70 epochs with a maximum of 3000 training steps.

## 9 DISCLOSURE OF LLM USAGE

We used a Large Language Model (LLM) to assist in enhancing the clarity and revising the writing of this manuscript. The contents are the sole responsibility and original work of the authors.

**Language Command**

**Locating Prompt**

Below is a Computer-Aided Design (CAD) operation sequence. Replace the parts that need to be modified with the string <mask> according to the editing instruction.

```
Original CAD Operation Sequence:

{
    "original_sequence": "circle,31,53,31,9,53,31,9,31 <curve_end>
        <loop_end> circle,31,37,31,25,37,31,25,31 <curve_end>
        <loop_end> <face_end> <sketch_end>
        add,31,63,31,31,31,1,0,0,0,0,1,0,-1,0,25,30,30 <extrude_end>"
}

Editing Instruction:

{
    "instruction": "Attach an annular groove around the central hole."
}

Masked CAD Operation Sequence:

{
    "masked_sequence": "circle,31,53,31,9,53,31,9,31 <curve_end>
        <loop_end> circle,31,37,31,25,37,31,25,31 <curve_end>
        <loop_end> <face_end> <sketch_end> <mask> <extrude_end>
        <mask>"
}
```

**Infilling Prompt**

Below is the original Computer-Aided Design (CAD) operation sequence.

```
Original CAD Operation Sequence:

{
    "original_sequence": "circle,31,53,31,9,53,31,9,31 <curve_end>
        <loop_end> circle,31,37,31,25,37,31,25,31 <curve_end>
        <loop_end> <face_end> <sketch_end>
        add,31,63,31,31,31,1,0,0,0,0,1,0,-1,0,25,30,30 <extrude_end>"
}
```

The parts that need to be modified according to the editing instruction have been replaced by the string <mask>.

```
Editing Instruction:

{
    "instruction": "Attach an annular groove around the central hole."
}

Masked CAD Operation Sequence:

{
    "masked_sequence": "circle,31,53,31,9,53,31,9,31 <curve_end>
        <loop_end> circle,31,37,31,25,37,31,25,31 <curve_end>
        <loop_end> <face_end> <sketch_end> <mask> <extrude_end>
        <mask>"
}
```

Based on the original CAD sequence, the editing instruction, and the masked sequence, generate the complete edited CAD sequence by replacing <mask> with the appropriate content.

Figure 4: An example of CAD editing instruction for both the locating and infilling stages.

**CAD editing using SCOPE**

**Locating Stage**
**(Input)**

"Attach an annular groove around the central hole."

```
{
    "original_sequence": "circle,31,53,31,9,53,31,9,31 <curve_end>
        <loop_end> circle,31,47,31,15,47,31,15,31 <curve_end>
        <loop_end> <face_end> <sketch_end>
        add,31,47,31,31,31,1,0,0,0,1,0,0,0,1,43,30,30 <extrude_end>",
}
```

**(Output)**

```
{
    "masked_sequence": "circle,31,53,31,9,53,31,9,31 <curve_end>
        <loop_end> circle,31,47,31,15,47,31,15,31 <curve_end>
        <loop_end> <face_end> <sketch_end>
        add,31,47,31,31,31,1,0,0,0,1,0,0,0,1,43,30,30 <extrude_end>
        <mask>"
}
```

**Infilling Stage**
**(Input)**

"Attach an annular groove around the central hole."

```
{
    "original_sequence": "circle,31,53,31,9,53,31,9,31 <curve_end>
        <loop_end> circle,31,47,31,15,47,31,15,31 <curve_end>
        <loop_end> <face_end> <sketch_end>
        add,31,47,31,31,31,1,0,0,0,1,0,0,0,1,43,30,30 <extrude_end>",

    "masked_sequence": "circle,31,53,31,9,53,31,9,31 <curve_end>
        <loop_end> circle,31,47,31,15,47,31,15,31 <curve_end>
        <loop_end> <face_end> <sketch_end>
        add,31,47,31,31,31,1,0,0,0,1,0,0,0,1,43,30,30 <extrude_end>
        <mask>"
}
```

**(Output)**

```
{
    "edited_sequence": "circle,31,53,31,9,53,31,9,31 <curve_end>
        <loop_end> circle,31,47,31,15,47,31,15,31 <curve_end>
        <loop_end> <face_end> <sketch_end>
        add,31,47,31,31,31,1,0,0,0,1,0,0,0,1,43,30,30 <extrude_end>
        circle,31,53,31,9,53,31,9,31 <curve_end> <loop_end>
        circle,31,48,31,14,48,31,14,31 <curve_end> <loop_end>
        <face_end> <sketch_end>
        add,31,39,31,31,31,1,0,0,0,1,0,0,0,1,63,30,30 <extrude_end>",
}
```

**(Text-to-CAD)**

Figure 5: End-to-end CAD editing using SCOPE.