# OpenReview forum: "SCOPE: Spatially-Constrained Parametric Editing for Text-Guided CAD Models"
_ICLR.cc/2026/Conference — ICLR 2026 Conference Withdrawn Submission_

### Official Review · Reviewer_VkAU · 2025-10-27

**Soundness:** 2
**Presentation:** 2
**Contribution:** 2
**Rating:** 2
**Confidence:** 4

**Summary:**

This works aims at improving the performance of the text-based CAD editing task.
Different to existing methods that lack explicit spatial understanding, it introduces SCOPE, which add special tokens called \<SR\> tokens to represent spatial relationships in both the locate and infill stage proposed by CAD-Editor (an existing framework).

**Strengths:**

- This work focuses on an interesting and practical task, i.e., text-based CAD editing.
- There are both quantitative and qualitative evaluations in the experiment.

**Weaknesses:**

- There are no ablation studies to demonstrate the effectiveness of the proposed techniques. First, there should be an experiment that \<SR\> tokens are not used. Current results of CAD-Editor cannot serve as that role as it use different backbone with different model size to SCOPE (e.g., Llama vs. Gemma and 8B vs. 12B). Second, there also should be an experiment that hierarchy-aware masking strategy (see lines 276-279) are not used.
- What is the scope and coverage of  \<SR\> tokens? For example, in the example in lines 257-265, will 'smaller' also be a  \<SR\> tokens? or  \<SR\> tokens are only related to relative position but not related to relative size?
- There is a very large overlap with CAD-Editor [1] in the writing. While I understand that CAD-Editor is a very important reference of this work, it should be very clear about which part is from CAD-Editor and which part is newly proposed by this work. In the current version, for audiences who have not read CAD-Editor, they may think all content of Section 3 are newly proposed by this work. However, the locate-infill stage in Section 3.1 and 3.2 is largely from CAD-Editor, the structured text representation in Section 3.3 is largely from FlexCAD [2], the data synthesis pipeline in Section 3.4 is largely from CAD-Editor.
- It would be better if there are more qualitative examples.

[1] https://arxiv.org/abs/2502.03997 (ICML'25)

[2] https://arxiv.org/pdf/2411.05823 (ICLR'25)

**Questions:**

see weaknesses

---

### Official Review · Reviewer_xHVn · 2025-10-29

**Soundness:** 3
**Presentation:** 3
**Contribution:** 3
**Rating:** 4
**Confidence:** 3

**Summary:**

The paper proposes SCOPE, a text-guided CAD editing framework that augments the “locate-then-infill” paradigm with an explicit spatial-relation token (<SR>) and a spatially-guided masking strategy. The authors also synthesize ~150K instruction–original–edited triplets with explicit spatial terms using an LVLM pipeline. On a DeepCAD-based benchmark, SCOPE reportedly improves alignment with spatial instructions, with its 12B backbone variant outperforming CAD-Editor on JSD, CD, and D-CLIP, while slightly trailing it on validity (VR).

**Strengths:**

1. The paper tackles a well-defined and practical limitation of current text-to-CAD systems. The inability to understand spatial instructions is a major hurdle for the practical use of such models in professional design workflows, which rely heavily on relative positioning.
2. The introduction of a dedicated <SR> token is an elegant and simple modification to the existing locate-then-infill framework. It provides a strong and explicit signal to the model, effectively transforming an implicit reasoning challenge into a more structured sequence modeling task.
3. The method achieves state-of-the-art performance, outperforming the CAD-Editor baseline it builds upon across most key metrics, including JSD (realism), CD (geometric distance), and D-CLIP (instruction-following).

**Weaknesses:**

1. Confounding Factors in Experimental Comparison: The main baseline, CAD-Editor, uses a Llama3-8B backbone, while SCOPE is evaluated on Gemma3-1B and Gemma3-12B. This makes a direct comparison difficult. While the authors show their 1B model outperforms the 8B baseline on D-CLIP, it performs significantly worse on all other metrics (VR, JSD, CD). It is therefore unclear how much of the SOTA 12B model's performance gain is attributable to the SCOPE methodology versus simply using a different and larger backbone model.
2. Lack of Ablation Study: The paper does not include an ablation study to isolate the impact of its two main contributions. The performance gain comes from both the new spatially-grounded dataset and the explicit <SR> token. A crucial missing experiment is training the model on the new dataset without the <SR> token. This would quantify the exact benefit of the explicit token, as it's possible that most of the gain comes from simply training on better, spatially-aware data.
3. Key claims lack direct metrics: The paper emphasizes better localization in the locate stage but reports no token-level or region-level localization metric (e.g., token F1/IoU on masks). Reliance on D-CLIP (image/text directional similarity) may not faithfully capture 3D spatial relations under varying views

**Questions:**

1. Can the authors explain the significant drop in the Validity Ratio (from 95.6% to 91.3%) compared to CAD-Editor? Why does adding spatial-constraint awareness make the model more likely to generate invalid geometry, and are there methods to mitigate this?
2. Given the different backbones used (Gemma3 vs. Llama3), how can the authors definitively attribute the performance gains to the SCOPE methodology rather than the choice of backbone?
3. Was an ablation study performed to measure the impact of the <SR> token separately from the spatially-grounded dataset? What is the D-CLIP score when training on the new dataset but without using the explicit <SR> token?
4. Why does the 12B model have lower VR than CAD-Editor? Any failure mode breakdown (parser errors vs geometric infeasibility)?

---

### Official Review · Reviewer_pgxE · 2025-11-01

**Soundness:** 2
**Presentation:** 2
**Contribution:** 2
**Rating:** 2
**Confidence:** 4

**Summary:**

This paper introduces SCOPE, a framework for text-guided CAD editing that augments the locate-then-infill paradigm with explicit spatial reasoning. The model integrates a new token for encoding spatial relations (e.g., above, left of) and a data-synthesis pipeline that generates spatially grounded instruction–model triplets. Through this integration, SCOPE aims to enhance localization accuracy and instruction-following for edits involving relative positioning or geometric constraints.

**Strengths:**

The paper addresses a clear gap in CAD editing—spatial understanding, which is essential for translating natural language instructions into geometry-aware modifications.

The paper is easy to follow.

**Weaknesses:**

The paper completely follows the locate-then-infill paradigm, which, to my knowledge, is a temporary compromise for addressing the challenges in CAD editing tasks. Introducing spatial relations into the task should focus on tackling more fundamental issues, such as improving LLM-based instruction following or refining spatial arrangements during CAD editing. Simply presenting a CAD-editor with additional spatial constraints makes it difficult for me to recognize the novelty of this framework.

This work employs too few metrics for CAD evaluation. For example, it does not assess visual quality using VLMs, which has become a recent trend in works such as Text2CAD [1] and CADFusion [3]. The F1-score from Text2CAD and the parametric accuracy from CAD-LLaMA [2] would also be useful starting points for evaluating editing quality. Evaluating CAD is a multi-perspective process, and the current evaluation scale seems insufficient to demonstrate state-of-the-art performance comprehensively—not to mention that the presented figures appear incremental to me.

The paper dedicates too many pages to describing previous works (~1.5 pages on CAD representations and the locate-and-infill framework, and ~1 page on defining metrics) but too few to qualitative demonstrations.

[1] Text2CAD: Generating Sequential CAD Models from Beginner-to-Expert Level Text Prompts. NeurIPS 2024.

[2] CAD-Llama: Leveraging Large Language Models for Computer-Aided Design Parametric 3D Model Generation. CVPR 2025.

[3] Text-to-CAD Generation Through Infusing Visual Feedback in Large Language Models. ICML 2025.

**Questions:**

How are directional cues (e.g., “left”, “above”) normalized across different camera angles—are they relative to a fixed CAD coordinate system or the rendered viewpoint?

Could the authors include more qualitative examples or visual-language evaluation metrics to substantiate spatial fidelity claims?

How would performance change if the model bypassed locate-then-infill and generated edits directly from instruction + input sequence?

---

### Official Review · Reviewer_TFnr · 2025-11-05

**Soundness:** 3
**Presentation:** 3
**Contribution:** 2
**Rating:** 2
**Confidence:** 3

**Summary:**

-   The paper introduces SCOPE, which explicitly injects spatial context via a dedicated $\<SR\>$ token by extending the locate-then-infill approach.
-   Furthermore, the authors propose an automated data synthesis pipeline to generate a large-scale spatially-grounded CAD editing dataset. This dataset contains 150k triplets: each consisting of an original model, a spatial instruction, and the corresponding edited model with relative position descriptors at all hierarchy levels.

**Strengths:**

- **[S1] Practical Problem:** Localized edits for CAD models is an important problem and has real-world applications. SCOPE proposes a practical approach to address the problem statement.
-   **[S2] Impressive Qualitative Results:** Figs. 2 and 3 show clear qualitative results demonstrating the  superior performance of the proposed method. The result in Fig. 2 for the prompt, “Add a cylinder with a central circular hole, aligning with the left hole,” is particularly impressive. Unlike other baseline methods, SCOPE correctly follows this input instruction.

**Weaknesses:**

-   **[W1] Validity Ratio (VR) Tradeoff:** In Tab. 1, VR scores for SCOPE (Gemma3-1B) and SCOPE (Gemma3-12B) are 75 and 91.3, respectively, less than CAD-Editor, which is 95.6.. Does that mean SCOPE produces many invalid sequences? This reduction in VR is a limitation and weakness of the proposed work.
-   **[W2] No Ablation studies:** SCOPE introduces the spatial relation $\<SR\>$ token into the vocabulary to extend locate-then-infill approaches. However, the paper lacks the following ablation experiments: 1.) SCOPE without $\mathcal{C}_{mask}$ and 2.) SCOPE without a hierarchy-aware masking strategy.
-   **[W3] Limited Novelty:** The authors extend the locate-then-infill approaches by turning spatial relations into explicit tokens ( $\<SR\>$) and combining that with hierarchy-aware masking. Further, the authors propose a data synthesis pipeline for spatially-grounded CAD editing datasets. These are valuable contributions but are incremental in nature.

**Questions:**

-   [Q1] Why is the VR score lower in comparison to CAD-Editor?
-   [Q2] Are there any failure scenarios of the proposed method?
-   [Q3] Is Chamfer Distance (CD) reported for the unchanged regions of the model? The CD improvement is marginal compared to the baseline method: CAD-Editor. It would help if the authors could provide a region-specific CD.

---

### Note · Authors · 2025-11-15

**Comment:**

We sincerely thank the reviewers and the area chairs for their time, feedback and constructive comments.

**Withdrawal Confirmation:**

I have read and agree with the venue's withdrawal policy on behalf of myself and my co-authors.